# New Treatments in Renal Cancer: The AhR Ligands

**DOI:** 10.3390/ijms21103551

**Published:** 2020-05-18

**Authors:** Boris Itkin, Alastair Breen, Lyudmila Turyanska, Eduardo Omar Sandes, Tracey D. Bradshaw, Andrea Irene Loaiza-Perez

**Affiliations:** 1Department of Oncology, Hospital General de Agudos Juan Fernandez, C1425 CABA Buenos Aires, Argentina; Borisitkin1@gmail.com; 2School of Pharmacy, Centre for Biomolecular Sciences, The University of Nottingham, University Park, Nottingham NG72RD, Nottinghamshire, UK; Alastair.Breen@lstmed.ac.uk (A.B.); tracey.bradshaw@nottingham.ac.uk (T.D.B.); 3Faculty of Engineering, University of Nottingham, University Park, Nottingham NG72RD, Nottinghamshire, UK; lyudmila.turyanska@nottingham.ac.uk; 4Facultad de Medicina, Instituto de Oncología Ángel H. Roffo (IOAHR), Universidad de Buenos Aires, Área Investigación, Av. San Martin 5481, C1417 DTB Buenos Aires, Argentina; eosandes@yahoo.com.ar

**Keywords:** renal cancer, AhR, aminoflavone, benzothiazoles, nanocompounds

## Abstract

Kidney cancer rapidly acquires resistance to antiangiogenic agents, such as sunitinib, developing an aggressive migratory phenotype (facilitated by c-Metsignal transduction). The Aryl hydrocarbon receptor (AhR) has recently been postulated as a molecular target for cancer treatment. Currently, there are two antitumor agent AhR ligands, with activity against renal cancer, that have been tested clinically: aminoflavone (AFP 464, NSC710464) and the benzothiazole (5F 203) prodrug Phortress. Our studies investigated the action of AFP 464, the aminoflavone pro-drug currently used in clinical trials, and 5F 203 on renal cancer cells, specifically examining their effects on cell cycle progression, apoptosis and cell migration. Both compounds caused cell cycle arrest and apoptosis but only 5F 203 potently inhibited the migration of TK-10, Caki-1 and SN12C cells as well as the migration signal transduction cascade, involving c-Met signaling, in TK-10 cells. Current investigations are focused on the development of nano-delivery vehicles, apoferritin-encapsulated benzothiazoles 5F 203 and GW610, for the treatment of renal cancer. These compounds have shown improved antitumor effects against TK-10 cells in vitro at lower concentrations compared with a naked agent.

## 1. Introduction

### 1.1. Renal Cancer Background

Renal cell carcinoma (RCC) is the sixth most common malignancy in men and tenth in women worldwide with a significant variation in incidence and mortality between different geographic regions. Incidence rates are highest in North America followed by Europe and Australia. The highest estimated mortality rates have been observed in Uruguay, Argentina, Chile and the USA [1]. Current histopathological classification of renal tumors totals more than ten histopathological and molecular subtypes of carcinomas. Clear cell renal cell carcinoma (ccRCC, near 75%), papillary renal cell carcinoma (type I and type II) and chromophobe renal cell carcinoma are the most common of them. Remaining subtypes are infrequent and account for less than 1% each [2].

Certain correlation between histopathologic subtypes and genomic alteration in the main RCC subtypes can be traced. Bi-allelic loss of function of the Von Hippel Lindau (VHL) tumor suppressor gene, as a consequence of mutations, deletions or epigenetic silencing, is found in the vast majority of ccRCC and is usually the most precocious truncal molecular driving event [2]. However, additional mutations, including PBRM1 (polibrome-1), SETD2 (SET Domain Containing 2), BAP1 (BRCA1 associated protein 1) and PI3K-AKT-mTOR, are usually needed for aggressive malignant phenotypes to arise. Type I papillary RCC is characterized by proto-oncogene c-Met over-expression/amplification as a result of activated mutations or copy number alterations [3,4]. It has recently been described that aryl hydrocarbon receptor (AhR) activation mediates kidney disease and renal cell carcinoma [5].

### 1.2. Current Therapies_ENREF_7_ENREF_4

At present, two categories of pharmacological treatment options for metastatic/advanced RCC are supported by relevant clinical practice guidelines: targeted therapies and immune therapies [6,7,8] _ENREF_9. The targeted therapies include vascular endothelial growth factor receptor tyrosine-kinase inhibitors (VEGFR TKIs) and rapalogues, which inhibit the mammalian target of the rapamycin (mTOR) pathway signaling. This therapy aims to override the effects of hypoxia-inducible growth factor activation. ”Classic” antiangiogenic agents (sorafenib, sunitinib, pazopanib, axitinib, bevacizumab) target the VEGF pathway, while more recently approved cabozantinib and lenvatinib also act upon the compensatory activation of putative resistance mechanisms, such as hepatocyte growth factor and fibroblast growth factor activation [3,4].

Immune check-point inhibitors (ICIs) are by far the most important subclass among the immunotherapy drugs. The use of cytokines interleukine-2 and interferon-alpha has largely declined [9]. Ipilimumab is a monoclonal antibody, which antagonizes CTLA-4 and selectively depletes T-regs in the tumor microenvironment, stimulating an anti-tumor immune response [10,11]. Monoclonal antibodies directed against programmed death receptor-1 (PD-1), e.g., nivolumab and pembrolizumab, and its ligand (PD-L1) Avelumab, inhibit the immunosuppressive co-signals mediated by PD-1/PD-L1 interaction, enhancing the anti-tumor activity of the T-cells [12].

Recently, several Phase 3 studies demonstrated that combinations of drugs are more efficacious compared to single agents, suggesting that combining agents with different mechanisms of action could offer a promising route to overcome tumor resistance and improve survival outcomes. However, overlapping toxicities have been encountered and merit further investigation [13,14,15]. Since added toxicities may limit the use of synergistic combinations in cancer treatments, there is an urgent demand for new active agents with a completely distinct mechanism(s) of action. In this review, we discuss the role of the AhR ligands aminoflavone and benzothiazoles as potential new agents for the treatment of advanced mRCC.

## 2. Aryl Hydrocarbon Receptor

The aryl hydrocarbon receptor (AhR) was initially identified as a receptor for environmental toxins, such as dioxin. It belongs to the helix–loop–helix transcription factor family. Other members of this family are the AhR nuclear translocator (ARNT), the SIM and PER proteins as well as hypoxia-inducible factor 1α(HIF 1α) [16,17,18,19]. AhR is a ligand-activated transcription factor. The first described ligands of AhR were exogenous: polycyclic and polyhalogenated hydrocarbons (benzopyrene, 3-methyl-colantrene), xenobiotics (phenobarbital) and other pesticides like tetrachlorodibenzo-*p*-dioxin (TCDD) [19].

AhR is localized within the cell cytosol constitutively, where it is part of an inactivated complex composed of heat-shock protein 90 (Hsp90) and a 43 kDa protein known as AIP (Figure 1). The role of Hsp90 involves a chaperone activity that keeps AhR in a ligand-binding configuration and prevents its nuclear translocation. Hydrophobic ligands of AhR enter the cell by diffusion and bind to the receptor associated with Hsp90. This event leads to dissociation of the cytoplasmic complex and to AhR nuclear translocation. Within the nucleus, AhR interacts with the aryl hydrocarbon nuclear transporter (ARNT) forming a heterodimer that binds to specific DNA sequences called xenobiotic response elements (XREs). This binding leads to the transcriptional activation of genes that possess these XREs in their promoter sequences. Some of the genes activated by AhR encode Phase I and II metabolic enzymes, such as cytochrome P450 (CYP) 1A1, CYP1A2 and CYP1B1. Transgenic mice with a constitutively active AhR spontaneously develop tumors and the repressor of the AhR (AHRR) functions as a tumor suppressor in multiple human tumors. The expression of this gene is activated by AhR. The AHHR is known to competitively dimerize with ARNT, repressing AhR activity. The expression of AAHR has been correlated with metastasis-free survival in breast cancers [20,21]. In addition, AhR forms as a Cul4B-based E3 ubiquitin ligase complex, inducing selective protein degradation. AhR regulation signaling can be controlled via nuclear export and subsequent AhR degradation through the ubiquitin–proteasome signaling pathway [22]. In addition to this canonical pathway, signaling through AhR can also be mediated through interactions with other regulatory proteins, such as estrogen. AhR interacts with multiple other signaling pathways and activates other cytosolic proteins, including β-catenin, Smads, ERK, p38MAPK, JNK, NF-κB and RB [23].

AhR activation was first described as a cellular response to promote elimination of ambient contaminants and xenobiotics [24,25,26]. In humans, AhR is localized in the liver, lungs, kidneys, placenta, lymphocytes, ovary and breast. AhR/ARNT complex activation is tissue-specific and depends on co-regulators being present in different cell types [25]. Only a few endogenous AhR ligands have been reported. The prostaglandins (e.g., prostaglandin G2) are of note weak AhR agonist. Bilirubin [27] and prostaglandin G2 [28] are considered atypical AhR ligands, lacking the usual small, aromatic, rigid compounds. 

Naturally occurring polyphenolic compounds, known as flavonoids (flavones, flavonols and isoflavones), are substrates of CYP1A1 and CYP1B1, and are present in vegetables, fruits, medicinal herbs, beverages and dietary supplements [29]. Flavonoid compounds represent the largest class of natural AhR ligands [30]. This is an interesting and promising field of research for nutraceuticals in antitumor therapy whose molecular target is AhR.

### 2.1. Aminoflavone and Benzothiazoles: AhR-Targeted Antitumor Therapies 

The flavone analogue aminoflavone (AF) (4H-1-benzopyran-4-one, 5-amino-2-(4-amino-3-fluorophenyl)-6,8-difluoro-7-methyl) (NSC 686288, KyowaHakko Kogyo Co., Ltd., Tokyo, Japan) (Figure 2) has emerged as a candidate for preclinical evaluation and advancement to Phase I clinical trials, based on demonstrable antitumor activity in mice bearing human tumor xenografts and in the United States National Cancer Institute anticancer drug screen. Human tumor cell lines that exhibited particular sensitivity to AF include those of breast and renal origin. AF treatment of sensitive human MCF-7 breast tumor cells revealed elevated *CYP1A1/1A2* protein levels [31]. Preclinically, the antitumor benzothiazoles 5F 203 and Phortress (Figure 2) evoked potent antiproliferative activity in breast and ovarian tumor models, inducing CYP1A1 expression and generating DNA adducts, which are converted to lethal strand breaks in sensitive cell lines and xenografts only [26,32]. 

### 2.2. Sensitivity of Renal Cell Carcinoma to Aminoflavone: Role of CYP1A1

In an effort to delineate cellular markers of sensitivity to AF in cells of renal origin, we performed investigations on established renal cell lines and a series of renal cell isolates from patients with confirmed clear cell and papillary renal disease. 

In vitro antiproliferative activity of AF was evaluated in the cell lines Caki-1, TK-10, A498,RXF-393, ACHN and SN12-C (National Cancer Institute (NCI) repository, NCI-Frederick, Frederick, Maryland), grown as described [33,34]. Briefly, for these studies cells were seeded into 24-well plates, allowed to grow for 48 h and treated with AF (10^−10^ to 10^−5^ M) for 72 h. Drug exposure was terminated by the addition of 50% trichloroacetic acid to a final 10% concentration. Cells were stained with sulforhodamine B and protein was determined spectrophotometrically. Values are shown as the mean ± SD of 10 preparations [35]. AF produced 100% (total) growth inhibition at sub-micromolar concentrations after 72 h exposure in 3 of the 6 renal cell lines used. Caki-1 was the renal cell line most sensitive to AF with the drug producing total growth inhibition at 90 nM. Two other cell lines, A498 and TK-10, were also sensitive to AF, with growth inhibition at AF concentrations of 200 and 400 nM, respectively. AF produced in vitro regression in each of these AF sensitive cell lines, as evidenced by the drug-induced loss of cellular protein during the treatment period. Three additional cell lines (ACHN, SN12-C and RXF-393) were judged AF resistant, since total growth inhibition was not achieved even at an AF concentration of 10 µM.

### 2.3. Effect of AF on Human Tumor Renal Xenografts

The Caki-1 human tumor xenograft was established as described [36,37]. Intraperitoneal (IP) and intravenous (IV) treatments were given on a QD X 5 schedule, beginning Day 13. AF treatment of mice bearing Caki-1 renal cell carcinoma produced 100% (6 of 6) tumor-free survivors at intraperitoneal 120, 80 and 53 mg/kg doses, and 2 of the 6 tumor-free survivors at 90 mg/kg intravenously. Values are reported as the mean ± SD in 20 vehicle controls and 6 animals per AF dose (Figure 3). In contrast, AF demonstrated negligible activity and produced no tumor-free survivors against the AF-resistant RXF-393 tumor (data not shown). It was noteworthy that a single course of treatment during 5 days had a lasting impact after a subsequent 6 to 7 weeks in the responsive Caki-1 model [33].

### 2.4. AF Sensitivity and Induction of CYP1A1 and CYP1B1 mRNA

AF induced *CYP1A1* and *CYP1B1* gene expression in human tumor renal cell lines. Human tumor renal cell lines were treated with 1 to 1000 nM AF for 24 h. RNA was isolated from the control and treated samples, and *CYP1A1* and *CYP1B1* gene expression was measured by real-time RT-PCR, as described [33]. Data are shown as the mean fold induction of the treated cells ± SD relative to the constitutive expression in the control cells in 7 samples from 2 independent experiments.

### 2.5. AF Induced Apoptosis in AF Sensitive Renal Tumor Cell Lines

AF induced apoptosis in AF sensitive human tumor renal cell lines. Apoptosis was quantified following exposure to 1 µM AF for 24 h using M30-Apoptosense kit, as described [33]. Values were represented as the mean ± SD of 3 preparations, as described [33].

AF treatment resulted in an over 10-fold increase in apoptosis in Caki-1 and A498, the two most sensitive renal cell lines, and an over 6-fold increase in apoptosis in TK-10, the other AF-sensitive renal cell line. In contrast, AF treatment of RXF-393 and ACHN, the two most AF-resistant cell lines, did not result in apoptosis induction.

### 2.6. AF Sensitivity and Covalent Binding in Human Renal Cell Strains

The preceding observations in continuous cell cultures prompted a parallel series of investigations using renal cell carcinoma isolates (termed renal cell strains) obtained from patients diagnosed with papillary and clear cell renal carcinoma undergoing therapeutic protocols at the NCI Urologic Oncology Branch. All patient-derived material was obtained after appropriate informed consent [33]. These cell isolates were used in an attempt to examine AF sensitivity in two different renal histological types with confirmed pathological diagnosis using cells that may more closely represent the human disease. A total of 13 isolates, from patients diagnosed with renal cell carcinoma (clear in 9 and papillary in 4) were examined for sensitivity to AF.

These renal cell strains were judged as AF-sensitive if GI_50_ was < 1 µM and AF resistant if GI_50_ > 1 µM. The AF concentration of 1 µM was chosen because it appeared to be achievable following intravenous administration of AF to preclinical toxicology animal subjects [33]. Seven of the nine renal cell strains of the clear cell histological type were AF resistant, of which five were very resistant to AF with a GI_50_ > 20 µM (Table 1). Two clear cell isolates were AF sensitive with GI_50_ values of 0.9 µM (cell strain 117) and 0.25 µM (cell strain 109). AF resistance of the majority of the renal cell strains of the clear cell histological type contrasted with the greater responsiveness of the papillary histological type to AF. Three of the four renal cell strains exhibiting the papillary histological type had a GI_50_ < 1 µM (Table 1). In the 13 cell strains, histology appeared to be somewhat predictive of the degree of AF sensitivity, as applied to GI_50_ values for the four papillary cells vs. the nine clear cell strains. Lower GI_50_ values were observed in papillary cell strains. 

Examination of covalent binding of AF metabolite(s) by these renal cell strains was examined. (^3^H)-AF (5-amino-2-(4-amino-3-fluorophenyl)-6,8-difluoro-7-methyl-((3-3H)-4H-1-benzopyran)-4-one) (specific activity 7.09 Ci/mMol) was used to treat renal tumor cell lines and renal cell strains. Cells were exposed to 5, 50 and 500 nM AF for 16 h at 37 °C in their respective culture medium. Radioactive medium was removed as described [33]. The varying degrees of sensitivity to AF indicated that most AF-sensitive cell strains had increased binding of the AF metabolite(s) compared to the AF-resistant cell strains (Table 1). In 13 cell strains, the degree of covalent binding appeared to be predictive of AF sensitivity, as applied to values for an AF concentration of 500 nM (1-sided Wilcoxon test *p* = 0.015). If one dichotomized the degree of binding (less than 40 K vs. greater than 40 K), high binding occurred in 4 cell lines: 3 of the 5 sensitive and 1 of the 8 resistant lines. This dichotomization appeared to be indicative of AF sensitivity (1-sided Fisher’s exact test applied to the 2 × 2 table, *p* = 0.11).

### 2.7. AF Sensitivity and Induction of CYP1A1 and CYP1B1 mRNA and Apoptosis in Human Tumor Renal Cell Strains

Induction of *CYP1A1* mRNA following the treatment of renal cell strains was observed in 4 of 5 AF-sensitive strains and in 4 of 8 AF-resistant strains. Induction of *CYP1B1* mRNA by AF was apparent in all AF-sensitive cell strains but in only 1 of 8 AF-resistant strains (see Table 1). Four renal cell strains, 2 clear cell (cell strains 117 and 171) and 2 papillary (112 and 146) histological types exhibited induction of CYP1A1 and CYP1B1 mRNA following treatment with AF. Three of these strains were sensitive. Most sensitive renal cell strains were of the papillary histological type.

Studies were also conducted to determine whether AF treatment of the strains exhibiting sensitivity to AF produces apoptosis, similar to that observed in AF-responsive human tumor renal cell lines. The five AF-sensitive cell strains demonstrated the highest levels of apoptosis, particularly the three most sensitive strains of the papillary histological type (146, 132 and 112) and the most AF-sensitive clear cell type (109). These four AF-sensitive renal cell strains, of which three were of the papillary histological type, were clearly more susceptible to apoptosis than the remaining nine renal cell strains. In the 13 cell strains the degree of apoptosis induction appeared to be predictive of AF sensitivity [33] (1-sided Wilcoxon test, *p* = 0.002). If one dichotomized the degree of apoptosis induction (less than 2-fold vs. greater than 2-fold), high induction occurred in four cell lines, of which all four were sensitive lines. This dichotomization appeared to be predictive of AF sensitivity (1-sided Fisher’s exact test applied to the 2 X 2 table, *p* = 0.01).

Sensitivity to AF, defined as GI_50_ < 1 µM, was correlated with covalent binding (dpm/ng cell protein), *CYP1A1* and *CYP1B1* mRNA expression (induction relative to control), as well as with apoptosis induction (relative to the control). All variables were measured at various AF concentrations. In the 13 renal cell strains, drug sensitivity was also correlated with tumor histology (papillary vs. clear cell). All results should be considered exploratory and hypothesis-generating rather than conclusive regardless of the *p*-values (which are provided for descriptive purposes and were not adjusted for multiple comparisons). In this context, only *p*-values < 0.10 should be considered suggestive of predictive ability for the molecular variable with respect to AF sensitivity. 

On the other hand, it should be noted that ability to detect predictive ability for the variables was severely limited by the small numbers of cell lines and cell strains. The tests used were nonparametric to avoid making assumptions concerning the statistical distribution of the variables or appropriate scale of measurement (the tests were invariant under monotonic scale transformations, such as transformation to logs of the original data). The primary test used was the Wilcoxon rank sum test (a nonparametric version of the more familiar test), which was used to relate AF sensitivity (a binary categorical variable) to the magnitudes of the various molecular variables (continuous variables) described. When appropriate, continuous variables were also dichotomized and related to AF sensitivity by Fisher’s exact test [38] (a nonparametric version of the more familiar chi-square test). Significance levels (*p*-values) are shown as 1- or 2-sided, as appropriate. All *p*-values were calculated by exact methods using StatXact-4 (Cytel Software Corp., Cambridge, MA, USA).

The results presented were directly related to early clinical evaluation of the agent. Our data were concordant with the possibility that tumor cells unable to metabolize the drug will not be growth inhibited. Hence, the assays described, including CYP1A1 induction following incubation with unlabeled drug or induction of apoptosis, could be used with tumor biopsies or fine needle aspirates to select patients with the greater likelihood of benefit, while avoiding the toxicity of drug administration in patients who have a low likelihood of benefit from the agent. These investigations provided evidence that sensitivity to AF can be assessed using a combination of covalent binding, induction of CYP1A1 and CYP1B1 mRNA, and susceptibility to apoptosis induction. Furthermore, these results suggest a trend toward differential sensitivity of renal cell carcinoma histological types to AF, with papillary renal cell carcinoma exhibiting greater AF sensitivity than clear cell renal cell carcinoma.

### 2.8. Aryl Hydrocarbon Receptor Activation by Aminoflavone: New Molecular Target for Renal Cancer Treatment

Considering that AF exhibits noteworthy evidence of antitumor activity in vitro and in vivo against neoplastic cells of renal origin, inducing CYP1A1, and the covalent binding of the AF-reactive intermediates and apoptosis, our research group evaluated the role of AhR, the main transcriptional regulator of CYP1A1, in the antiproliferative effects of AF in human renal cancer cells. AF-cytotoxicity in human renal cell lines and a renal cancer cell strain was assessed by MTS assay in the presence or absence of an AhR inhibitor. Drug-induced AhR nuclear translocation was evaluated by Western blotting of AhR in cytosolic and nuclear fractions and by measuring xenobiotic response element-driven luciferase activity. Apoptosis induced by the drug was evaluated by 4,6-diamidino-2-phenylindole and acridine orange/ethidium bromide staining and by measuring phosphorylated P53 (*p*-P53) and P21 levels, caspase 3 activation and poly(ADP-ribose) polymerase cleavage. AF inhibited cell growth in a dose-dependent manner in TK-10, Caki-1, SN12-C and A498 human renal cells, but not in ACHN cells.

The antiproliferative effect of AF was abrogated by preincubation of TK-10, Caki-1 and SN12-C cells with the AhR antagonist, α-naphthoflavone. AF treatment also induced apoptosis in TK-10, Caki-1 and SN12-C cells, which was not observed in ACHN cells. AF induced time-dependent AhR nuclear translocation and AhR transcriptional activity in sensitive renal cancer cell lines. A renal cell strain derived from a human papillary tumor also showed sensitivity to AF, as well as AhR pathway activation and drug-induced apoptosis. We concluded that AhR translocation could be included as a marker of sensitivity to AF in sensitive renal tumor cells of different histological origin, in Phase II clinical trials.

Statistical significance between three or more groups was calculated by one-way analysis of variance (ANOVA) followed by Tukey’s test. To compare two groups, the unpaired Student’s t-test with a Welch correction was used. Statistical analysis was performed using GraphPad InStat version 3.06 for Windows 95 (GraphPad Software Inc., San Diego, CA, USA; www.graphpad.com). Designations for statistical significance were * *p* < 0.05 and ** *p* < 0.01 [39] (Figure 4).

### 2.9. In Vitro Antitumor Effects of AhR Ligand Benzothiazole (5F 203) in Human Renal Cell Carcinoma

Phortress, the lysylamide prodrug of 2-(4-amino-3-methylphenyl)-5-fluorobenzothiazole (5F 203), underwent Phase I clinical evaluation and achieved disease stabilization in 33% of the patients recruited [40]. Interestingly, preclinical NCI 60 cell line panel data showed that TK-10 cells and other human renal cancer cell lines were consistently sensitive to 5F 203 and Phortress [41]. Intriguingly, Phortress stabilized disease in the two renal carcinoma patients recruited to the trial (in one patient, stability was maintained for 16 cycles). Therefore, we sought to examine AhR pathway activation and CYP1A1 inducibility in TK-10 and other renal carcinoma cell lines after treatment with 5F 203. Given the poor prognosis associated with kidney cancer and the paucity of therapeutic options, preclinical investigations of the use of aminophenylbenzothiazole experimental antitumor agents against these tumors are warranted. 

### 2.10. AFP 464 and 5F 203 Altered Cell Cycle Distribution and Evoked Apoptosis in Sensitive Renal Cancer Cells

As previous results indicate that AFP 464 (NSC710404) induced DNA damage and apoptosis in renal cancer cells [39] and 5F 203 evoked DNA damage in breast and ovarian cancer cells [34,42], we investigated perturbations in the cell cycle after treatment of renal cells with these compounds. For this approach, cells were exposed to 1 µM AFP 464, 1 µM 5F 203 or 0.1%DMSO for 24 and 48 h and subsequently processed for cell cycle analyses. AFP 464 only caused an increase in phase G0/G1 in TK-10 cells at 24 h. In contrast, the ACHN cell cycle was not perturbed following treatment with AFP 464. 5F 203 caused an increase in phase G0/G1 in SN12C and Caki-1 and TK-10 cells. In contrast, the ACHN cell cycle was not perturbed following treatment with 1 µM 5F 203.

A total of 20,000 events were analyzed, and one representative experiment, of three, is shown. The values represented the average of three independent experiments; * *p* < 0.05 and ** *p* < 0.01 with respect to the control cells [43].

### 2.11. Impact of AFP 464 and 5F 203 Treatments on Cell Migration

We investigated the effect of AFP 464 and 5F 203 treatments on migration of renal tumor cells in vitro by wound healing assay. AFP 464 treatment decreased migration neither in the sensitive nor in the resistant cell lines. In contrast, 5F 203 (1 µM) significantly suppressed cell migration in the three sensitive cell lines, namely, the TK-10, SN12C and Caki-1 cell lines, respectively. For the non-sensitive cell line (ACHN), migration was not affected. The AFP 464 wound healing assay was performed as follows: TK-10, SN12C and ACHN cells were incubated with AFP 464 for 24 h with the dilutions described in the methodology. A wound was made with a yellow tip; the initial and final wound areas were measured (Tf = 20 h post wound) and analyzed with the ImageJ program. The graph shows the percentage migration for each dilution realized. Values that are significantly different from the controls are indicated (* *p* < 0.05). Representative fields from one experiment are shown under each graph. Experiments were performed in triplicate. For the 5F 203 wound healing assay, TK-10, SN12C, Caki-1 and ACHN cells were incubated with 5F 203 for 24 h with the dilutions described in the methodology. The initial and final wound areas were measured (Tf = 14 h post wound) and analyzed with the Image J program. The graph shows the percentage migration for each dilution realized. Values that are significantly different from the controls are indicated (* *p* < 0.05). Representative fields from one experiment are shown under each graph. Experiments were performed in triplicate [43].

### 2.12. Effect of 5F 203 on c-Metphosphorylation in TK-10 Cells

As treatment with 5F 203 demonstrated inhibition of TK-10 cell migration, and *p*-Met is involved in the migration process, cell lysates were subjected to *p*-Met Western blot analyses. TK-10 cells were exposed to 1 μM 5F 203. Compared to the control, c-Met phosphorylation was downregulated at all time points examined. Lysates were prepared on three separate occasions; representative phospho c-Met (pMet), total c-Met (Met) and GAPDH (loading control) blots are shown. Densitometry was performed on all blots to quantify c-Met, phosphorylated c-Met and GAPDH expression. The pMet/Met, pMet/GAPDH and Met/GAPDH (D) ratios were calculated. The values represented the average of three independent experiments; * *p* < 0.05 and ** *p* < 0.01 with respect to the control cells.

We concluded that AhR ligand antitumor agents, such as AFP 464 and 5F 203, represent potential new candidates for the treatment of renal cancer. Both compounds caused cell cycle arrest and apoptosis. 5F 203 is sequestered by TK-10 cells and induces CYP1A1 expression; 5F 203 potently inhibited the migration of TK-10, Caki-1, and SN12C cells, and inhibited c-Met receptor phosphorylation in TK-10 cells. C-Met receptor signal transduction promotes migration and metastasis. Therefore, we consider that 5F 203 offers potential for the treatment of metastatic renal carcinoma [43] (Figure 5).

### 2.13. Development of Benzothiazole Nanocopounds for Renal Cancer Treatment

As previously described, antitumor benzothiazoles (exemplified by 5F 203, and also 2-(3,4-dimethoxyphenyl)-5-fluorobenzothiazole (GW 610)) demonstrate potent and selective anticancer activity in vitro and in vivo. As potent AhR ligands, these planar molecules trigger activation of AhR signal transduction, resulting in their bioactivation by cytochromes P450 (CYPs) 1A1 and 2W1 [44]. The electrophilic products lead to lethal DNA-adduct generation and cancer cell death. The renal cell line TK-10 has demonstrated sensitivity to this class of antitumor agent and represents an intractable cancer phenotype for which prognoses are poor. Despite promising activity, the high lipophilicity and poor aqueous solubility of the benzothiazoles limit their antitumor clinical application. The apoferritin (AFt) protein cage has been proposed as a robust drug delivery vehicle; it exhibits stability, is biocompatible, species-specific, non-immunogenic and biodegradable. The AFt capsule is comprised of 24 polypeptide units that self-assemble into a spherical protein cage; the capsule possesses channels (0.3–0.4 nm), allowing transport of metal ions, small metal complexes and organic molecules, and is of uniform size (12 nm with an internal cavity of 8 nm that naturally stores ≤4500 ferric ions). AFt, as a drug delivery vehicle, can lead to increased delivery of the drug to the cancer tissue via exploitation of the enhanced permeation and retention (EPR) associated with the tumor microenvironment. AFt is recognized by transferrin receptor-1 (TfR-1; upregulated in cancer cells that exhibit high iron demand) and internalized into cells via clathrin-coated pit-mediated endocytosis. It is trafficked through the endosome system, and the pH-dependent disassembly of the AFt capsule in the acidic lysosomes allows controlled release of the cargo. Thus, we proposed that AFt may represent a good antitumor benzothiazole delivery vehicle, enhancing aqueous solubility and bioavailability, promoting the accumulation of the antitumor agent at the tumor sites, potentially improving efficacy and decreasing potential adverse reactions.

Encapsulation of antitumor benzothiazoles was exhaustively optimized using 5F 203 (Figure 6); [45]. Methods explored included pH- and urea-mediated reassembly and the nanoreactor, diffusion method. Optimization led to adoption of the nanoreactor method, for AFt encapsulation of 5F 203 and all subsequent benzothiazoles. HEPES buffer was used and the equivalent of 100 molecules of 5F 203 was added gradually at pH 5.5. In this method, loss of protein and its precipitation were minimized and encapsulation of ~70 molecules of 5F 203 per AFt cage was achieved (Table 2). Dynamic light scattering (DLS) and transmission electron microscopy (TEM; Figure 6b) confirmed that neither AFt size (13.1 ± 1.6 nm) nor capsule integrity had been affected by entrapment of 5F 203. Time-of-flight secondary ion mass spectrometry (ToF-SIMS) was performed, confirming that 5F 203 had been encapsulated within the AFt interior and was not merely AFt-surface-bound (Figure 6c). A series of distinctive peaks corresponding to AFt (56–58 *m*/*z*) and 5F 203 (255–295 *m*/*z*) were detected. With increasing sputter time, the 25–58 *m*/*z* peak intensity diminished and was accompanied by an increased signal in the 255–295 *m*/*z* 5F 203 range. Crucially, the 5F 203 and AFt peaks were not concomitantly visible: depth profiling using an argon cluster beam revealed 5F 203 peaks exclusively below the AFt surface, within the cavity. Anionic residues on the AFt interior surface bestow on it a net negative charge. To enhance the encapsulation efficiency of 5F 203, Phortress was encapsulated within AFt. Adopting the optimized nanoreactor method, ~130 Phortress molecules per AFt cage were achieved. Release of both 5F 203 and Phortress AFt capsules was complete within 24 h.

The structurally related dimethoxyphenylbenzothiazole GW 610 (Figure 7) also exhibits potent and selective antitumor activity against the TK-10 renal cell line. However, its poor pharmaceutical properties limit applications. Thus, to enhance aqueous solubility and bioavailability, GW 610 was encapsulated within AFt protein cages [46]; using the nanoreactor method, ~190 molecules of GW 610 per cage were entrapped. A series of amino acid ester prodrugs was synthesized to examine the effect of polarity and charge on encapsulation. During CYP-catalyzed biotransformation, GW 610 initially undergoes demethylation to yield 5-fluoro-2-(4-hydroxy-3-methoxyphenyl)benzothiazole (GW 608). GW 608’s exposed 4-hydroxy group allows conjugation of amino acids via an ester link.

Herein, we focus upon the lysyl–ester GW 608 conjugate. Whereas ~110 molecules of GW 608 were encapsulated per AFt cage, >380 molecules of GW 608-Lys per AFt cage were entrapped. We concluded that (i) the increased polarity of GW 608 (compared to GW 610) hinders encapsulation within the negatively charged cavity; and (ii) charge also impacts encapsulation, and (as in the case for Phortress) conjugation with positively-charged lysine (GW 608-Lys) resulted in enhanced encapsulation compared to GW 610, GW 608 and the serine (polar), glycine (non-polar) and aspartic acid (-ve) ester conjugates. At physiologically relevant temperature (~37 °C), 100% release of GW 610 and GW 608-Lys was observed within 12 h. All encapsulated benzothiazoles were stable at 4 °C over a period of >3 months with respect to drug-loading, as quantified by UV–vis spectroscopy.

To understand the effect of encapsulation on in vitro antitumour activity of the benzothiazoles, MTT and clonogenic assays were conducted. In MTT assays, TK10 cells were exposed to the test agent for 72 h before growth and the test agent growth inhibitory/cytotoxicity were assessed; in clonogenic assays, clonal survival was assessed following exposure of the cells to the test agents for 24 h. Thereafter, the test agent was removed, the cells washed and the medium replenished. Colonies were allowed to form until they reached a size of ≥50 cells in the control wells. The results of the MTT assays are summarized in Table 2, and representative dose response curves following exposure of the TK10 cells to the naked- and AFt-encapsulated GW 610 and Phortress is shown in Figure 8a,c. The AFt vehicle alone had no effect on cell growth or clonal survival. However, AFt encapsulation of benzothiazole molecules enhanced their potency against TK10 carcinoma cells: AFt-5F 203 and AFt-GW 610 were ~4-fold more potent; lysyl-derivatives Phortress and GW 608-Lys demonstrated a ~60-fold enhanced activity following AFt encapsulation. These results were corroborated in clonogenic assays where the AFt-encapsulated agent inhibited TK-10 survival and colony formation to a greater extent than naked benzothiazole, as exemplified in the Figure 8c inset. Phortress (administered at GI_50_ value—6.3 µM) inhibited colony formation by ~80%, whereas AFt-Phortress (0.1 µM) completely abolished clonal survival. Both naked- and AFt-GW610 similarly abolished colony formation when administered at GI_50_ values (Figure 8a). Importantly, we can conclude that tumor selectivity displayed by 5F 203, GW 610 and their amino acid prodrugs is maintained following AFt encapsulation. Neither GW 610 nor AFt-GW 610 inhibited growth of benzothiazole-insensitive MRC-5 fibroblasts.

We tested the hypothesis that Aft encapsulation may promote more rapid uptake of benzothiazoles in sensitive tumor cells. 5F 203 and GW 610 rely upon their lipophilic nature to diffuse across cell membranes before being recognized by cytosolic AhR. The more polar nature of the amino acid prodrugs may hinder this process, leading to reduced perceived activity compared to parent agents (as observed in Table 2). However, AFt is sequestered into cells via transferrin recetor-1 (TfR-1)-mediated endocytosis; TfR-1 is upregulated and highly expressed by cancer cells, including TK-10 [46]. TK-10 cells were seeded (2.5 × 10^5^) in 12-well plates and allowed 24 h to attach before being treated with naked- or AFt-benzothiazole. Following 3 h, 6 h or 9 h exposure periods, cells were harvested and cellular uptake (of benzothiazole) was determined by flow cytometry. It was evident that AFt encapsulation potentiated the TK-10 cellular uptake of the benzothiazoles, as exemplified in Figure 9a. After 6 h and 9 h exposure, intracellular levels of GW 608-Lys were significantly greater (*p* < 0.01) following treatment of cells with AFt-GW 608-Lys (compared to naked agent). Intracellular retention of the benzothiazoles in the benzothiazole-sensitive carcinoma cells is made possible by cytosolic AhR, leading to AhR-signal transduction activation. Indeed, it has been shown that AFt encapsulation does not affect cytochrome P450 (CYP) 1A1 induction by 5F 203 [45]. This result is in accordance with the observed release of benzothiazoles from their AFt cages. In insensitive MRC-5 fibroblasts where TfR-1expression is below detection limits [47] and CYP 1A1 expression is neither constitutive nor inducible, negligible uptake of benzothiazole molecules is observed, whether delivered naked or AFt encapsulated (Figure 9b). This observation is consistent with the selective and potent in vitro antitumor activity of AFt-encapsulated benzothiazoles.

## 3. Discussion

Previous studies in our research groups have shown that 5F 203 and AF caused growth inhibition in renal cells, which was accompanied by CYP1A1 induction and apoptosis in sensitive cells. Both compounds caused cell cycle arrest and apoptosis but only 5F 203 potently inhibited migration of TK-10, Caki-1 and SN12C cells and the migration signal transduction cascade, involving c-Met, in TK-10 cells. In our studies, we used a panel of renal cancer cell lines with different von Hippel Lindau (VHL) statuses [39]. Our results indicated that sensitivity or resistance to AF is independent of VHL status and HIF expression. This is significant as it widens the variety of renal tumors that can be effectively treated with this agent. Recently, it has been demonstrated that AF inhibits HIF1α expression in an AhR-independent fashion in certain breast, renal and ovarian cancer cell lines [48].

In contrast to AFP 464, 5F 203 significantly decreased cell migration in the three sensitive cell lines. A decrease in wound healing ability was observed in sensitive cell lines, compared to the control. Consistent with loss of migratory potential, a significant decrease in c-Met phosphorylation was observed at 1 h in TK-10 cells treated with 1 µM 5F 203. Inhibition of c-Met activity by 5F 203 is consistent with previous observations: 5F 203 (1 µM; 24 h) decreased c-Met phosphorylation by 85% and 69 % in MCF-7 and MDA-MB-435 breast carcinoma cells, respectively [49].

We speculate that 5F 203, a potent AhR ligand, triggers activation of a signaling cascade that potentially inhibits HIF signal transduction and hence the subsequent c-Met activation [49].

However, HIF1α expression after treatment with 5F 203 has to be measured in future studies. Met signal transduction is a key pathway for the treatment of renal cancer [50] and is also involved in metastasis progression; therefore, we consider that 5F 203 has potential for the treatment of metastatic renal carcinoma.

Treatment of sensitive cells with AF caused the translocation of AhR to the nucleus and the induction of AhR transcriptional activity. In addition, experiments were performed with the renal cell strain 112, which was derived from a papillary tumor sensitive to AF. These cells also showed AhR activation by AF, which was in agreement with CYP1A1 induction previously observed in this renal cell strain. Additionally, we observed AhR nuclear translocation after treatment with the drug. In our previous report, we indicated that papillary renal tumors are more sensitive to AF than clear cell tumors [33]. The enhanced activity of AF against the papillary variant of renal cell carcinoma is of special value. Except for temsirolimus and sunitinib (both have proven their efficacy in the treatment of non-clear cell kidney cancer and are recommended for clinical use), there are little or no data regarding the safety and efficacy of the new target drugs in papillary histology and there is a need for the development of new effective therapies. However, AhR activation by AF has to be confirmed in future studies using other papillary and clear cell carcinoma tumors. In contrast to the other cell lines, ACHN cells showed resistance to AF treatment, which was associated with the lack of induction of CYP1A1 and CYP1B1 transcription. We demonstrated that AhR activation does not occur in these cells in response to AF. We hypothesize that this may lead to the lack of activation of CYP1A1 with the consequent lack of ability of these cells to metabolize AF. In ACHN cells treated with DMSO only, AhR was present in the cytosol and nucleus. After treatment with 1 μM AF, for 0.5 to 6 h, immunoreactive AhR protein levels do not change in the cytosolic and nuclear fractions. Of note, we observed high levels of AhR in the nucleus, even in the control, indicating that AhR may be constitutively nuclear in this cell line. This is a common pattern of AhR subcellular distribution in cell lines of different origin resistant to AF [33]. A suitable hypothesis may be that nuclear export or degradation signals may be altered, leading to inappropriate AhR recycling. However, this has to be tested in future studies. A different AhR variant present in the ACHN cells could explain the fact that AF does not induce translocation of the receptor in this cell line. Differential expression of proteins that regulate AhR activation, such as HSP90, may also induce resistance to AF [33].

Since our data are concordant with the thesis that tumor cells that metabolize the drug will exhibit sensitivity to AF or 5F 203, the assays described, including CYP1A1 induction following incubation with unlabeled drug or induction of apoptosis, could potentially be performed ex vivo on tumor biopsies or fine needle aspirates to guide selection of patients with the greatest prospect of treatment benefit. Such selection would prevent potential toxicity associated with drug administration in patients who have a low likelihood of treatment benefit. However, according to the current landscape of early drug development, this may not be a clinical practice reality.

AhR was predominantly expressed in the nuclei of high-grade clear cell RCC (ccRCC) and tumor-infiltrating lymphocytes (TILs), and its expression levels in cancer cells and TILs correlated with the pathological tumor stage and histological grade. A multivariate Cox analysis revealed that the strong expression of AhR in cancer cells was a significant and independent predictor of disease-specific survival. AhR ligands upregulated the expression of AhR and CYPs and promoted invasion by upregulating the MMPs. Furthermore, AhR siRNA downregulated CYPs, and inhibited cancer cell invasion together with the downregulation of the MMPs. These results suggest that AhR regulates the invasion of ccRCC and may be involved in tumor immunity. Therefore, inhibiting the activation of AhR may represent a potentially attractive therapeutic target for ccRCC patients [51].

Regarding the AhR and cell cycle progression, it has also been reported that activation of the AhR by tranilast [52], an antiallergy medication that has demonstrated AhR agonistic activity, resulted in inhibition of mammosphere formation in drug-surviving cancer stem cells in the triple-negative breast cancer cell line MDA-MB-231, as well other oncogenic cell lines (BT474, SUM149 and SUM159). Tranilast also disrupted the interaction of the AhR with CDK4, resulting in cell cycle arrest.

The indole-3-carbinol metabolite, diindoilmetano (DIM), has also demonstrated antitumor activity mediated by AhR acting on tumor stem cells [53].

On the other hand, omeprazole (OM) is an AhR agonist and a proton-pump inhibitor that is used to treat people with gastric acid-related diseases [54]. OM as an AhR ligand depends not only on their structure but also on target organs and downstream reactions and genes. In the classical nuclear AhR/ARNT-mediated reaction, OM recruited AhR into the region of the c-x-c chemokine receptor 4 (CXCR4) promoter containing XRE, which was accompanied by loss of pol II on the promoter and decreased expression of CXCR4. Cancer cell CXCR4 overexpression contributes to tumor growth, invasion, angiogenesis, metastasis, relapse and therapeutic resistance. Therefore, omeprazole inhibits tumor invasion and regulates metabolism in vivo by inhibiting CXCR4 transcription [55].

Our group has demonstrated that the AF and AF pro-drug (AFP464) disrupt the mammospheres derived from breast cancer cells and a M05 mammary mouse model of breast cancer, respectively. We further examined the capacity of AF and AFP464 to exhibit anticancer activity and modulate the expression of the “stemness” genes, including α6-integrin. AF disrupted the mammospheres and prevented secondary mammosphere formation. In contrast, AF did not disrupt the mammospheres derived from AhR ligand-unresponsive MCF-7 cells. AFP464 treatment suppressed M05 tumor growth and disrupted the corresponding mammospheres. AF andAFP464 reduced the expression and percentage of cells that stained for “stemness” markers, including α6-integrin in vitro and in vivo, respectively. These data suggested AFP464 thwarts bulk breast tumor and TIC growth via AhR agonist-mediated α6-integrin inhibition [56]. In future studies, it will be important to evaluate the action of AF and 5F 203 on the tumor stem cell population of renal tumors. Further studies to evaluate the action of other AhR ligands, such as Tranilast, DIM or omeprazole, on renal cancer stem cells, also should be performed.

Considering that AhR has recently emerged as a physiological regulator of the innate and adaptive immune responses, in a previous publication we described that AFP 464 modulates the immune response in the estrogen-dependent, Tamoxifen-sensitive spontaneous M05 mammary carcinoma model. Splenic cells and tumor inflammatory infiltrates were studied by cytometric analyses. The modulation of splenocytes cytotoxic activity by AFP 464 was also evaluated. We further investigated the effects of AFP 464 on peritoneal macrophages by evaluating metalloproteinase, arginase and iNOS activities. We found that AFP 464 increased splenic cytotoxic activity, diminished the number of systemic and local Treg lymphocytes and MDSCs as well as and induced a M1 phenotype in peritoneal macrophages of M05 tumor-bearing mice. Therefore, we conclude that AFP 464 modulates immune responses, which collaborates with its anti-tumor activity [57]. Our results place the immune system as a novel target for this anti-tumor agent. Considering these data, future studies need to elucidate the possible AF immunomodulation against renal tumors. The effect of 5F 203 on the immune system in these tumors also should be evaluated. Considering the great impact of the immunomodulators on the treatment of renal cancer, combination of these AhR ligands with immunotherapy would be an interesting option to pursue in future studies.

The use of nano-scale drug delivery vehicles in renal cancer also offers an interesting option in cancer treatment, minimizing toxicity. Our results showed that antitumor benzothiazoles, such as 5F 203 and GW 610, potently inhibit the growth of certain renal carcinoma cells in vitro. These benzothiazoles are themselves prodrugs, efficiently bioactivated by CYP 1A1 and CYP 2W1 enzymes, whose inducible or constitutive expression is found in some cancers, including renal cell carcinoma. AFt encapsulation of 5F 203, GW 610 or their lysine-derivatives provides a robust formulation that enhances aqueous solubility (and bioavailability), increases intracellular uptake and significantly augments in vitro antitumor potency, whilst maintaining excellent tumor selectivity. Further in vivo preclinical evaluation is justified.

Considering that the expression of AhR target genes in the blood, including *CYP1A1* and *AhRR*, is also upregulated in CKD patients compared to healthy controls [58], in future studies not only fine needle aspirates from kidney tumors but also blood samples could be used to measure CYP1A1 levels, as a sensitivity marker, in patients treated with AF or 5F 203. In addition to CYP1A1 induction and AhR nuclear translocation, other markers of sensitivity to 5F 203 have been identified in ovarian tumors. 5F203 induced enhanced CYP1A1 expression; AhR translocation and reactive species formation (ROS) in IGROV-1 cells and ascites-isolated ovarian cancer cells that were sensitive to 5F203. In IGROV-1 cells, 5F203-induced ROS formation was accompanied by JNK, ERK and P38MAPK phosphorylation, as well as DNA damage and cell cycle arrest prior to apoptosis. In contrast, 5F203 failed to induce CYP1A1 expression, AhR translocation or oxidative stress in 5F203-resistant SKOV-3 cells, or in ovarian cancer ascites cells inherently resistant to this agent. We developed a bioassay, to measure the putative biomarkers of sensitivity to this agent that have been proposed using ascites of ovarian cancer patients [42]. In future studies it could be of interest to measure ROS formation and JNK, ERK and P38MAPK phosphorylation in renal cancer cells treated ex vivo with 5F 203 as a putative additional biomarker of renal tumors suitable to be treated with this antitumor agent.

Considering that AhR activation mediates kidney disease and renal cell carcinoma, and that declining renal function leads to the retention of various metabolites [5] that contributes to a variety of diseases, especially chronic kidney disease (CKD) and cardiovascular disease (CVD), as well as that uremic toxins from tryptophan metabolites that activate AhR contribute to these diseases, uremic toxins may provide new potential therapeutic approaches targeting AhR activation. Therefore, further studies should be performed on the effect of novel metabolites on AhR activity. AF and 5F 203 may only represent a fraction of the AhR ligands possessing potential therapeutic effects in renal cancer. Other metabolites with therapeutic actions derived from nutraceuticals, such as flavonoids, may be found in the near future.

In conclusion, AF and 5F 203 represent experimental anticancer agents that target novel molecular targets pertinent to renal cell carcinoma: AhR, CYP 1A1 and CYP 2W1. The mechanism of action, distinct from current clinical pharmacopeia, offers pharmacodynamics biomarkers of sensitivity that could guide patient selection.

## Figures and Tables

**Figure 1 ijms-21-03551-f001:**
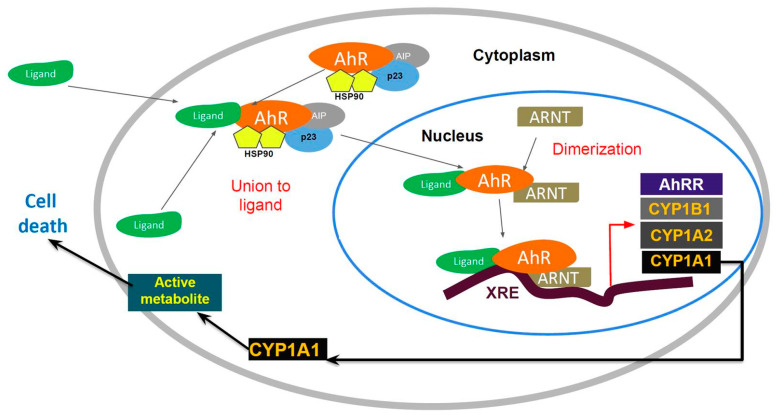
The aryl hydrocarbon receptor (AhR) signaling pathway.

**Figure 2 ijms-21-03551-f002:**
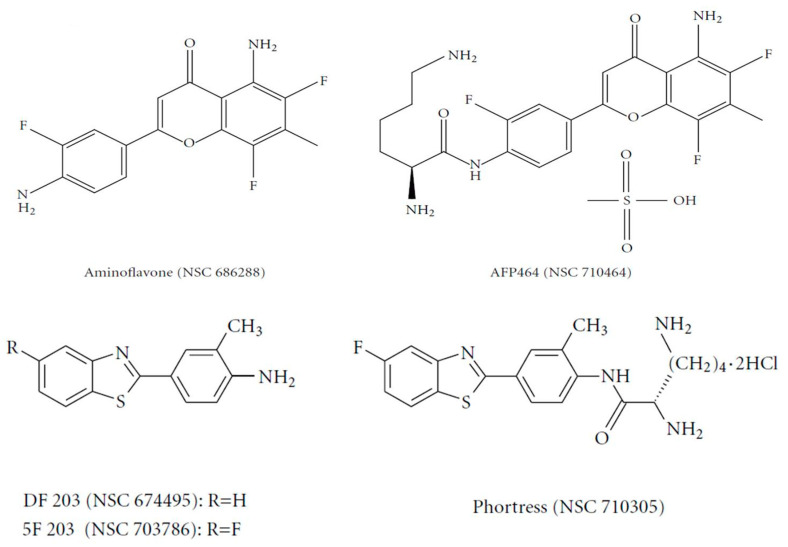
Chemical structures of aminoflavones and benzothiazoles.

**Figure 3 ijms-21-03551-f003:**
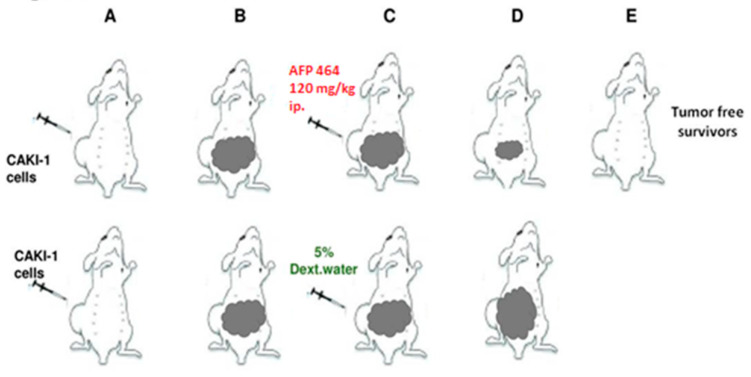
In vivo antitumor activity of aminoflavone (AF) against a Caki-1 human renal tumor (Figure reproduced from [33]).

**Figure 4 ijms-21-03551-f004:**
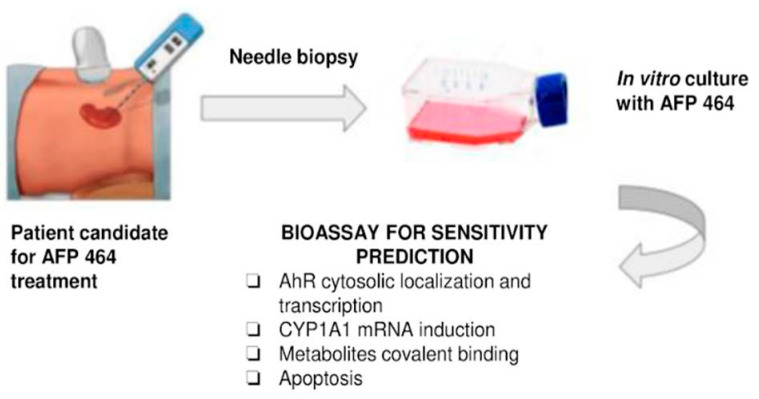
Proposed procedure for identification of patients whose renal tumors may be responsive to AFP 464 treatment (data reported in [33,39]).

**Figure 5 ijms-21-03551-f005:**
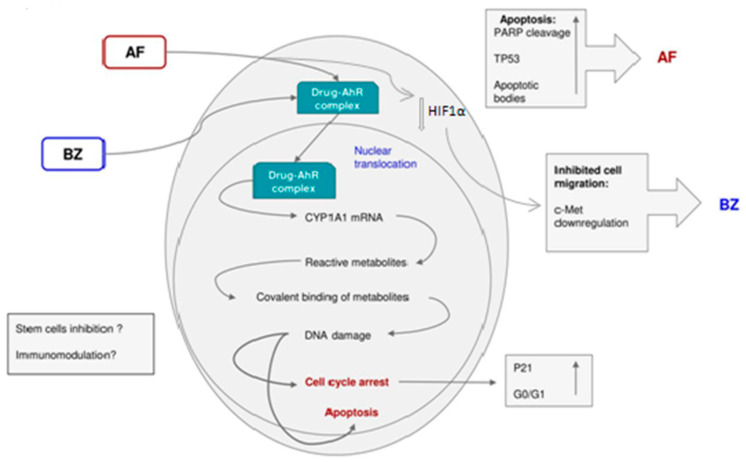
Proposed Mechanism of action for the AhR ligands AF and 5F 203 in renal cancer cells (AF, aminoflavone; BZ, benzothiazole 5F 203).

**Figure 6 ijms-21-03551-f006:**
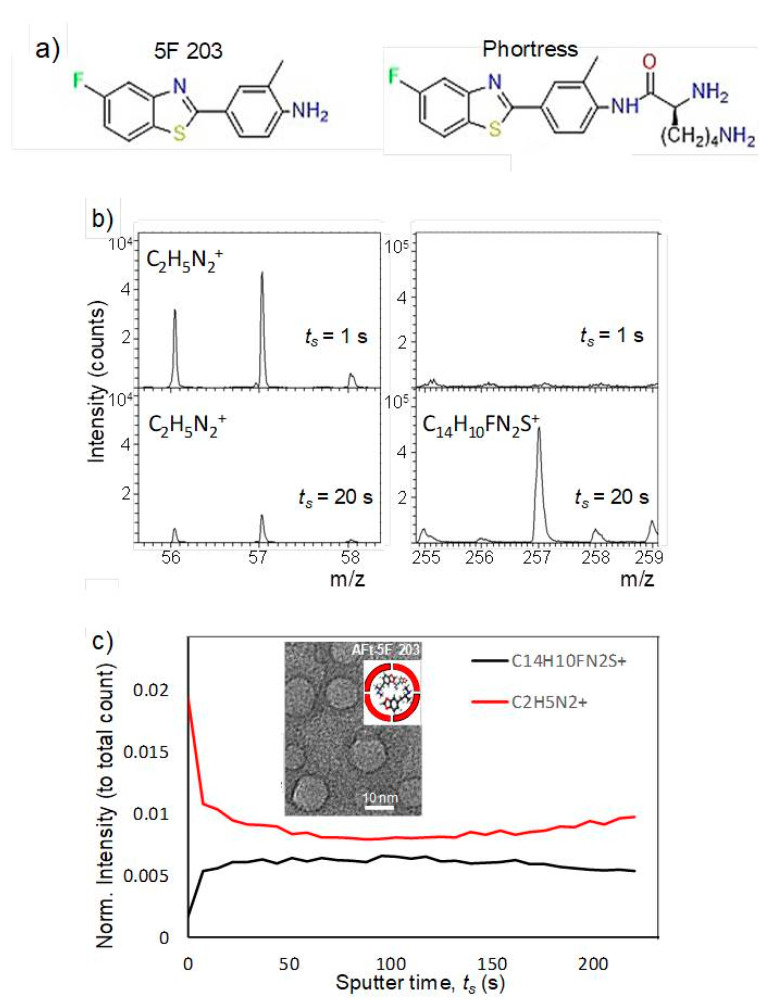
(**a**) Chemical structures of 5F 203 and Phortress. (**b**) ToF-SIMS mass spectra of AFt-5F 203. (**c**) Summary of intensity change with increasing sputter time. Inset: Transmission electron microscopy image of AFt-encapsulated 5F 203 and cartoon representation of AFt-5F 203 (figure reproduced from [46]).

**Figure 7 ijms-21-03551-f007:**
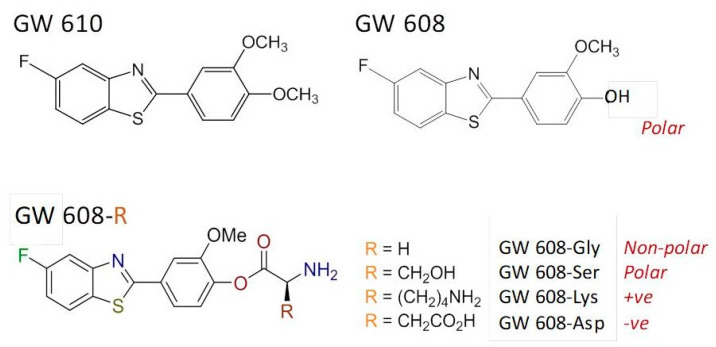
Chemical structures of the GW 610, GW 608 and amino acid prodrugs.

**Figure 8 ijms-21-03551-f008:**
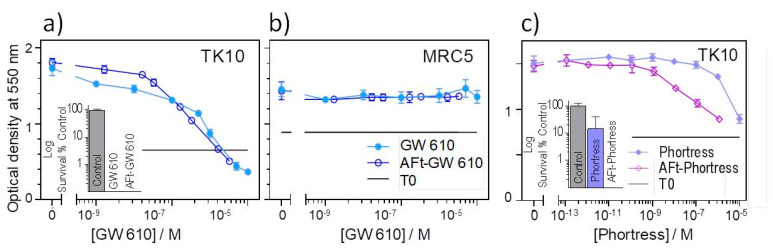
Representative growth inhibitory dose-response profiles following exposure of (**a**) and (**c**) TK10 carcinoma cells and (**b**) MRC-5 fibroblasts (72 h) to naked- or AFt-encapsulated (**a**) and (**b**) GW610 or (**c**) Phortress (points are means ± SD, *n* = 4). Inset in (**a**) and inset in (**c**): TK10 clonal survival following 24 h exposure of cells to naked and AFt-encapsulated GW610 with GI_50_ values. Data points are the mean ± SD of 3 independent trials (*n* = 3 per trial).

**Figure 9 ijms-21-03551-f009:**
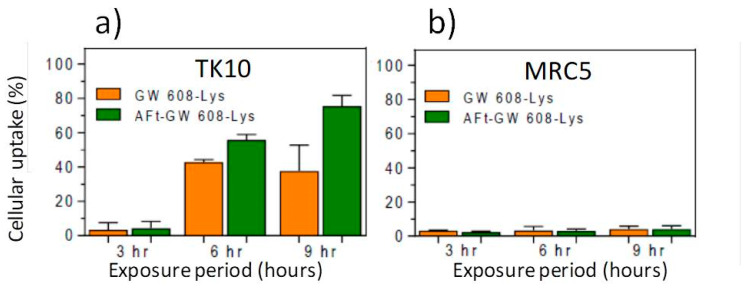
Cellular uptake of GW 608-Lys and AFt-GW 608-Lys by benzothiazole-sensitive TK10 carcinoma cells (**a**) and -insensitive MRC-5 fibroblasts (**b**). Data points are the mean ± SD of 3 independent trials (*n* = 3 per trial).

**Table 1 ijms-21-03551-t001:** Renal cell strains were exposed to 0, 1, 10, 100 and 1000 nM AF for 24 h in Dulbecco’s modified Eagle’s medium containing 4.5 gm/L D-glucose supplemented with 5% fetal calf serum. Cells were processed, and CYP1A1 and CYP1B1 mRNA was quantitated by RT-PCR. Fold CYP1A1 or CYP1B1 mRNA induction relative to the untreated controls is indicated regarding the AF concentration for which induction was observed. Cells with high apoptosis are shown (data reported in [33]).

		AF nM Concentration (Fold mRNA Induction)		
**AF resistant**	**Type**	***CYP1A1* mRNA**	***CYP1B1* mRNA**	**Highest apoptosis**
115	Clear	None	None	
121	Clear	100 (4), 1000 (82)	None	
150	Clear	*None*	*None*	
151	Clear	None	None	
154	Clear	100 (2)	None	
171	Clear	100 (2), 1000 (10)	100 (2); 1000 (7)	
181	Clear	None	None	
124	Papillary	1, 10, 100, 1000 (2)	None	
**AF sensitive**				
109	Clear	None	1 (4); 10 (2)	x
117	Clear	100 (2), 1000 (7)	100 (2); 1000 (5)	
112	Papillary	10 (2), 100 (11), 1000 (27)	1, 10 (2); 100 (3); 1000 (7)	x
132	Papillary	1, 100, 1000 (2), 10 (9)	1000 (2)	x
146	Papillary	1, 10, 100, 1000 (2)	1 (8); 10 (27); 100 (39); 1000 (27)	x

**Table 2 ijms-21-03551-t002:** A tabulated summary of GI_50_ values following treatment of TK10 renal carcinoma cells with naked- and AFt-encapsulated antitumor benzothiazoles. MTT assays were conducted following 72 h exposure of cells to test agent. *n* = 3 per trial; ≥3 independent trials. The mean number of benzothiazole molecules encapsulated per AFt cage is also depicted.

Benzothiazole	GI_50_ Value (µM) ± SD		No. Molecules: AFt Cage ± SD
	Naked Agent	AFt-Encapsulated	
5F 203	0.11 ± 0.09	0.028 ± 0.004	71
Phortress	6.3 ± 2.9	0.098 ± 0.008	130
GW 610	0.57 ± 0.16	0.14 ± 0.09	191
GW 608-Lys	34.1 ± 14	0.6 ± 0.35	386

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
