# Peer review of "New Treatments in Renal Cancer: The AhR Ligands"

_ijms, 2020, doi:10.3390/ijms21103551_

Round 1
Reviewer 1 Report
I would like to congratulate the authors for their own scientific work presented in this review. The review summarizes the very promising results of a new possible therapeutic approach for the treatment of renal cell carcinoma.
With a little regret, however, I see that some results have not been statistically analyzed to adequately prove them. The importance of statistical data analysis is crucial in the preparation of experiments, so that a difference or no difference can also be proven statistically. This is a well-known problem. Looking ahead, it will certainly be very interesting to read further publications on this topic by the authors.
The reference list should be revised, please. In line 614/615 the information is incomplete.
Author Response
-I would like to congratulate the authors for their own scientific work presented in this review. The review summarizes the very promising results of a new possible therapeutic approach for the treatment of renal cell carcinoma.
Thank you very much for your compliments about our work. We do hope that our results could show a new possible therapeutic approach for the treatment of renal cell carcinoma.
-With a little regret, however, I see that some results have not been statistically analyzed to adequately prove them. The importance of statistical data analysis is crucial in the preparation of experiments, so that a difference or no difference can also be proven statistically. This is a well-known problem. Looking ahead, it will certainly be very interesting to read further publications on this topic by the authors.
We want to let you know that the entire review was completely re-written showing the statistically analyses of all data. Please read the new version submitted. We also revised the figure legends and Figures 3 and 5 were modified.
-The reference list should be revised, please. In line 614/615 the information is incomplete
The reference list was revised. More references were added according with the changes in the text.

Reviewer 2 Report
In the present work, the authors provided a mix of a review and original work to dissect the possible role of AhR ligands in the treatment of kidney cancer and their pharmacotherapy delivery.
According to this reviewer, the authors should consider a specific editorial format to present the data. While interesting, data are presented with single statements and all the laboratory experiments are showed in images and tables – with possibility to reference a certain experiment.
A detailed methodology sections must be provided.
While the role of AhR in kidney tumorigenesis and kidney cancer has been reported, the authors reduce all the cancerogenesis mediated by AhR to the CYP- induction. As a transcription factor related to the response to xenobiotics, the cellular reprogramming in response to the ligands is quite extensive. For instance, the antiproliferative activity of one agonist ligand, aminoflavone, has been related in other experiences to the inhibition of HIF-1alpha expression in an AhR-independent fashion – that has been excluded from the models presented by the authors. In general, the exact pathogenesis is addressed quite superficially and the biological activities reported in detail up to the transcriptional release of CYP1A1. For the critical role of HIF-1alpha in renal cancerogenesis, the assumptions require some speculations on the network of AhR, and the implications on the model.
The authors screened cellular model known to be sensitive or resistant to AhR ligands, and assess the biologic activity by assessing the levels of one transcriptionally- regulated target of the xenobiotic response. While this is interesting, again the pathogenetic mechanism is not reported. Moreover, a simple assessment of the ligand and the cellular localization (nucleus, cytoplasm) at the baseline, as predictors of the AhR activity, is not reported- not differences between clear cell and non- clear cell histology addressed. For instance, the finding in papillary- type deserves a better understanding.
The authors should underline the speculative nature of some considerations, arguing on the clinical feasibility of functional treatment- response assays on xeno- models. For instance, “results presented were directly related to early clinical evaluation of the agent. Since our 221 data were concordant with the possibility that tumor cells that cannot metabolize the drug 222 will not be growth inhibited, the assays described, including CYP1A1 induction following 223 incubation with unlabeled drug or induction of apoptosis, could potentially be used with 224 tumor biopsies or fine needle aspirates to select patients with the greatest likelihood of 225 benefit, while avoiding the toxicity of drug administration in patients who have a low 226 likelihood of benefit from the agent” appears less likely to be a clinical practice reality – according to the current landscape of early drug development.
More data must be provided on the anticancer effects of AhR ligands – the authors basically reported the effects in cancer cells, mirroring the transcriptional patterns in the non-cancer settings.
Author Response
-In the present work, the authors provided a mix of a review and original work to dissect the possible role of AhR ligands in the treatment of kidney cancer and their pharmacotherapy delivery.
According to this reviewer, the authors should consider a specific editorial format to present the data. While interesting, data are presented with single statements and all the laboratory experiments are showed in images and tables – with possibility to reference a certain experiment.
A detailed methodology sections must be provided.
Thank you for your comments. We found them very interesting. We want to let you know that the entire review was completely re-written showing the methodology used for all experiments and the statistically analyses of all data. Please read the new version submitted. We also revised the figure legends and Figures 3 and 6 were modified. The editorial format chosen is a review, were we present our own work and the work of other authors.
While the role of AhR in kidney tumorigenesis and kidney cancer has been reported, the authors reduce all the cancerogenesis mediated by AhR to the CYP- induction. As a transcription factor related to the response to xenobiotics, the cellular reprogramming in response to the ligands is quite extensive. For instance, the antiproliferative activity of one agonist ligand, aminoflavone, has been related in other experiences to the inhibition of HIF-1alpha expression in an AhR-independent fashion – that has been excluded from the models presented by the authors. In general, the exact pathogenesis is addressed quite superficially and the biological activities reported in detail up to the transcriptional release of CYP1A1. For the critical role of HIF-1alpha in renal cancerogenesis, the assumptions require some speculations on the network of AhR, and the implications on the model.
This is a very good observation. We agree with you, the role of HIF1alpha is critical in renal carcinogenesis. For that reason agonist ligand, aminoflavone, has been related in other experiences to the inhibition of HIF-1alpha expression in an AhR-independent fashion. We included this data in the discussion section (first and second paragraphs).
The authors screened cellular model known to be sensitive or resistant to AhR ligands, and assess the biologic activity by assessing the levels of one transcriptionally- regulated target of the xenobiotic response. While this is interesting, again the pathogenetic mechanism is not reported. Moreover, a simple assessment of the ligand and the cellular localization (nucleus, cytoplasm) at the baseline, as predictors of the AhR activity, is not reported- not differences between clear cell and non- clear cell histology addressed. For instance, the finding in papillary- type deserves a better understanding.
Treatment with AF caused the translocation of AhR to the nucleus and the induction of AhR transcriptional activity. In addition, experiments were performed with the renal cell strain 112, which was derived from a papillary tumor sensitive to AF. These cells also showed AhR activation by AF, which was in agreement with CYP1A1 induction previously observed in this renal cell strain.
Additionally we observed AhR nuclear translocation after treatment with the drug. In our previous report, we indicated that papillary renal tumors are more sensitive to AF than clear cell tumors (Loaiza Perez et al, 2004). The enhanced activity of AF against the papillary variant of renal cell carcinoma is of special value. Except for temsirolimus and sunitinib, (both have proven their efficacy in the treatment of non-clear cell kidney cancer and are recomended for clinical use), there are litlle or no data regarding the safety and efficacy of the new target drugs in papillary histology and there is a need for the development of new effective therapies . However, AhR activation by AF has to be confirmed in future studies using other papillary and clear cell carcinoma tumors. In contrast to the other cell lines, ACHN cells showed resistance to AF treatment which was associated with the lack of induction of CYP1A1 and CYP1B1 transcription. We demonstrated that AhR activation does not occur in these cells in response to AF . We hypothesize that this may lead to the lack of activation of CYP1A1 with the consequent lack of ability of these cells to metabolize AF. In ACHN cells treated with DMSO only,
AhR was present in the cytosol and nucleus. After treatment with 1 μM AF, for 0.5 to 6 h, immunoreactive AhR protein levels do not change in the cytosolic and nuclear fractions. Of note, we observed high levels of AhR in the nucleus even in the control, indicating that AhR may be constitutively nuclear in this cell line. This is a common pattern of AhR subcellular distribution in cell lines of different origin resistant to AF (Loaiza Perez et al, 2004).
A suitable hypothesis may be that nuclear export or degradation signals may be altered leading to inappropriate AhR recycling. However, this has to be tested in future studies. A different AhR variant present in ACHN cells could explain the fact that AF does not induce translocation of the receptor in this cell line. Also, differential expression of proteins that regulate AhR activation, such as HSP90, may induce resistance to A (Loaiza Perez et al, 2004).
.
The authors should underline the speculative nature of some considerations, arguing on the clinical feasibility of functional treatment- response assays on xeno- models. For instance, “results presented were directly related to early clinical evaluation of the agent. Since our 221 data were concordant with the possibility that tumor cells that cannot metabolize the drug 222 will not be growth inhibited, the assays described, including CYP1A1 induction following 223 incubation with unlabeled drug or induction of apoptosis, could potentially be used with 224 tumor biopsies or fine needle aspirates to select patients with the greatest likelihood of 225 benefit, while avoiding the toxicity of drug administration in patients who have a low 226 likelihood of benefit from the agent” appears less likely to be a clinical practice reality – according to the current landscape of early drug development.
This paragraph was added in the discussion:
Since our data were concordant with the possibility that tumor cells that cannot metabolize the drug will not be growth inhibited, the assays described, including CYP1A1 induction following incubation with unlabeled drug or induction of apoptosis, could potentially be used with tumor biopsies or fine needle aspirates to select patients with the greatest likelihood of benefit, while avoiding the toxicity of drug administration in patients who have a low likelihood of benefit from the agent” appears less likely to be a clinical practice reality – according to the current landscape of early drug development
More data must be provided on the anticancer effects of AhR ligands – the authors basically reported the effects in cancer cells, mirroring the transcriptional patterns in the non-cancer settings.
Data regarding other AhR ligands like Omeprazol, DIM, Tranilast and naturally occurring flavonoids were added in the discussion.
Round 2
Reviewer 2 Report
Dear authors,
the review has been partially edited. The changes have not been all marked in the final draft, so this reviewer tried to understand the changes provided, and they are:
- a small paragraph, in the conclusion
- some specifications on the methodology.
Please make sure the images used, and extracted from other works - though with a reference - have been obtained with permission.
Author Response
“The review has been partially edited. The changes have not been all marked in the final draft, so this reviewer tried to understand the changes provided, and they are:
- a small paragraph, in the conclusion
- some specifications on the methodology.
Please make sure the images used, and extracted from other works - though with a reference - have been obtained with permission.
Thank you for your comments. The final version of the review has been done (Please read the new version). It includes some changes (marked in yellow) such as:
English corrections, a small paragraph in the conclusion.
Please note that some references with methodology specifications have been added to the list. For example:
- Skehan, P., Storeng, R., Scudiero, D., Monks, A., McMahon, J., Vistica, D. et al: New colorimetric cytotoxicity assay for anticancer-drug screening. J Natl Cancer Inst, 82: 1107, 1990
- Plowman, J., Dykes, D. J., Hollingshead, M., Simpson-Herren, L. and Alley, M. C.: Human tumor xenograft models in NCI drug development. In: Anticancer Drug Development Guide: Preclinical Screening, Clinical Trials, and Approval. Edited by B. Teicher. Totowa, New Jersey: Humana Press Inc., pp. 101–125, 1997
- Geran, R. I., Greenberg, N. H., Macdonald, M. M., Schumacher, A. M. and Abbott, B. J.: Protocols for screening chemical agents and natural products against animal tumors and other biological systems. Cancer Chemother Rep, 3: 51, 1972
- Snedecor, G. W. and Cochran, W. G.: Statistical Methods, 7th ed. Ames, Iowa: The Iowa State University Press, 1982
Regarding the Figures:
We prepared schemes and Tables using our own data. They are not the same figures of other papers, we reproduced data of our publications. Therefore, we do not attach permissions.
